# Exploring the behavioral determinants of COVID-19 vaccine acceptance among an urban population in Bangladesh: Implications for behavior change interventions

**Md. Abul Kalam**[1]***, Thomas P. Davis, Jr.**[2], **Shahanaj Shano**[3,4], **Md. Nasir Uddin**[3], **Md. Ariful Islam**[5], **Robert Kanwagi**[2], **Ariful Islam**[4], **Mohammad Mahmudul Hassan**[6], **Heidi J. Larson**[7,8]

**1** Bangladesh Country Office, Helen Keller International, Dhaka, Bangladesh, **2** World Vision International, Geneva, Switzerland, **3** Institute of Epidemiology, Disease Control and Research (IEDCR), Dhaka, Bangladesh, **4** EcoHealth Alliance, New York, New York, United States of America, **5** Bangladesh Country Office, SNV Netherlands Development Organisation, Dhaka, Bangladesh, **6** Chittagong Veterinary and Animal Science University, Chattogram, Bangladesh, **7** Vaccine Confidence Project, London School of Hygiene and Tropical Medicine, London, United Kingdom, **8** Institute for Health Metrics and Evaluation, University of Washington, Seattle, Washington, United States of America

* a.kalam724@gmail.com

**Data Availability Statement:** All relevant data are within the paper and its Supporting Information file 2 (S2 File: Tabulation sheet with data).

## Abstract

### Background

While vaccines ensure individual protection against COVID-19 infection, delay in receipt or refusal of vaccines will have both individual and community impacts. The behavioral factors of vaccine hesitancy or refusal are a crucial dimension that need to be understood in order to design appropriate interventions. The aim of this study was to explore the behavioral determinants of COVID-19 vaccine acceptance and to provide recommendations to increase the acceptance and uptake of COVID-19 vaccines in Bangladesh.

### Methods

We employed a Barrier Analysis (BA) approach to examine twelve potential behavioral determinants (drawn from the Health Belief Model [HBM] and Theory of Reasoned Action [TRA]) of intended vaccine acceptance. We conducted 45 interviews with those who intended to take the vaccine (Acceptors) and another 45 interviews with those who did not have that intention (Non-acceptors). We performed data analysis to find statistically significant differences and to identify which beliefs were most highly associated with acceptance and non-acceptance with COVID-19 vaccines.

### Results

The behavioral determinants associated with COVID-19 vaccine acceptance in Dhaka included perceived social norms, perceived safety of COVID-19 vaccines and trust in them, perceived risk/susceptibility, perceived self-efficacy, perceived positive and negative

**Funding:** This study was partially funded by Bangladesh Bureau of Educational Information and Statistics (BANBEIS), Grant #SD2019967. Mohammad Mahmudul Hassan has been supported through this grant.

**Competing interests:** The authors have declared that no competing interests exist.

consequences, perceived action efficacy, perceived severity of COVID-19, access, and perceived divine will. In line with the HBM, beliefs about the disease itself were highly predictive of vaccine acceptance, and some of the strongest statistically-significant (p<0.001) predictors of vaccine acceptance in this population are beliefs around both injunctive and descriptive social norms. Specifically, Acceptors were 3.2 times more likely to say they would be very likely to get a COVID-19 vaccine if a doctor or nurse recommended it, twice as likely to say that most people they know will get a vaccine, and 1.3 times more likely to say that most close family and friends will get a vaccine. The perceived safety of vaccines was found to be important since Non-acceptors were 1.8 times more likely to say that COVID-19 vaccines are "not safe at all". Beliefs about one's risk of getting COVID-19 disease and the severity of it were predictive of being a vaccine acceptor: Acceptors were 1.4 times more likely to say that it was very likely that someone in their household would get COVID-19, 1.3 times more likely to say that they were very concerned about getting COVID-19, and 1.3 times more likely to say that it would be very serious if someone in their household contracted COVID-19. Other responses of Acceptors on what makes immunization easier may be helpful in programming to boost acceptance, such as providing vaccination through government health facilities, schools, and kiosks, and having vaccinators maintain proper COVID-19 health and safety protocols.

## Conclusion

An effective behavior change strategy for COVID-19 vaccines uptake will need to address multiple beliefs and behavioral determinants, reducing barriers and leveraging enablers identified in this study. National plans for promoting COVID-19 vaccination should address the barriers, enablers, and behavioral determinants found in this study in order to maximize the impact on COVID-19 vaccination acceptance.

## Introduction

As of 18 June 2021, there have been more than 177 million cases of severe acute respiratory syndrome coronavirus 2 (SARS-CoV-2), or COVID-19, and more than 3.8 million deaths have resulted from the pandemic in 224 countries [1]. The pandemic poses an immense threat to the global public health system and is causing social and cultural disruptions [2, 3]. Bangladesh reported its first clinically-confirmed COVID-19 case on March 8, 2020, and as of 14 June 2021, Bangladesh has 826,922 COVID-19 confirmed cases with 13,118 deaths [4]. The social and economic costs of COVID-19 have been significant in the developing countries like Bangladesh [5]. As the pandemic is expected to continue to impose enormous burdens of morbidity and mortality, and to severely disrupt societies and economies, the administration of effective COVID-19 vaccines is the only clinical preventive measure [6]. As of 14 June 2021, Bangladesh has administered 10,072,344 doses of Oxford/AstraZeneca (COVISHIELD) and BIBP (SinoPhrama) vaccines in the whole country, out of which, 5,822,177 people have received their 1st dose, and 4,250,167 have completed the two-dose schedule [4]. However, like many other countries, the government has initially focused vaccination administration on very specific groups of people with the general public expected to be eligible for vaccines at a later date.

Vaccine hesitancy or refusal around COVID-19 vaccines is a growing concern worldwide, especially as new and deadly variants emerge. The World Health Organization (WHO) identified vaccine hesitancy as one of the top ten global health threats in 2019 [7] and hesitancy has been a problem in COVID-19 vaccination, as well. A multi-country survey found that only 71.5% of participants reported that they would be very or somewhat likely to get a COVID-19 vaccine [8]. A rapid systematic review of 23 peer-reviewed studies and 103 additional syndicated surveys around COVID-19 vaccine hesitancy in the US and globally showed that perceived risk, concerns over vaccine safety and effectiveness, doctors' recommendations, and inoculation history were common influencing factors for vaccine hesitancy [9]. COVID-19 vaccine hesitancy was found to be growing between March and November 2020 [10], but improvements in vaccine acceptance have been noted in some parts of the world since November [11]. In Bangladesh, one study reported around 81% of urban (metropolitan, district and municipalities) people showed willingness to be vaccinated [12]. However, this study did not report at what point in time (e.g., immediately, within six months) people were willing to be vaccinated. Several studies reported a higher hesitancy or delay (around 32–42 percent) in metropolitan areas of Bangladesh [13, 14] which is aligned with one nationally representative survey [15]. Furthermore, more than a quarter of the population in Bangladesh are under 18 years of age [16] who were not yet eligible for COVID-19 vaccination at the time of this study. Considering this, herd immunity will be even more difficult to achieve without very high vaccine acceptance among adults once a vaccine is available to a large portion of the population. While COVID-19 vaccines ensure increased individual protection, with hesitancy and delay in getting vaccines present among a higher proportion of people, it is critical to explore the reasons for hesitancy in order to prevent community transmission [17]. Since vaccine hesitancy for COVID-19 vaccines is relatively high, it is critical to explore the behavioral factors influencing it.

A growing number of studies have identified demographic, socioeconomic, and behavioral factors that are linked with vaccine acceptance. These factors include age and marital status [18, 19], level of education and ethnic origin [19–21], previous vaccination with the influenza vaccine [20, 22], and gender [18]. Moreover, mistrust, misconceptions, misinformation, and lack of knowledge among community members on vaccine-preventable diseases have been considered influential determinants of lower levels of acceptance [8, 9, 20, 23, 24]. These factors have influenced vaccine uptake during previous pandemics and outbreaks caused by H1N1, MERS, SARS, and Ebola virus [25–28].

A meta-analysis demonstrated that the use of behavioral change models (e.g. the HBM and Theory of Planned Behavior) would be useful for identifying the influencing determinants of vaccine acceptance [23]. The use of economic models when studying vaccine acceptance or hesitancy exhibit some shortcomings in describing the determinants [23, 29, 30]. Behavioral studies have shown that the decision to vaccinate is often based on perceived benefits, effectiveness, and perceived risk of vaccine side-effects versus infection [31]. Systematic reviews on behavioral determinants of health have shown that the HBM was useful in identifying determinants associated with the acceptance of Human Papillomavirus (HPV) [29] and influenza vaccination uptake [32]. This model has also been found to be effective in predicting intention to vaccinate against influenza among health care workers in Jordan [33]. Similarly, a study using Theory of Planned Behavior (TPB) (which was developed from the TRA) showed that vaccine intentions were determined by attitudes, subjective norms, and perceived behavioral control regarding vaccinations among college students [34]. In a comparative study of TPB and TRA, it was found that attitudes and perceptions of social support were determinants for HPV vaccination uptake [30].

Vaccine hesitancy and acceptance are complex in nature, and vaccine decisions can vary according to context, time and place [35]. Global studies on demographic determinants can have limited value when looking at determinants of COVID-19 vaccine acceptance in a given country, time, and geographical area. The Technical Advisory Group on Behavioral Insights and Sciences for Health of WHO has identified a number of behavioral drivers including enabling environment, social influences and motivation, and recommended to contextualize these drivers into national plans of COVID-19 vaccination [36]. Understanding how different behavioral attributes affect individual preferences about vaccination at as granular level as feasible can help inform public health authorities about the actionable activities and messages that will be necessary to achieve broader community uptake of vaccines. Therefore, the primary objective of this study was to explore the behavioral determinants of COVID-19 vaccine acceptance among people of different income levels in urban communities in Bangladesh. The secondary objective of this study is to provide policy recommendations of culturally-acceptable behavioral change intervention points to address these determinants in order to improve COVID-19 vaccine uptake.

## Materials and methods

### Study site and context

This Barrier Analysis study was conducted in different urban areas of Dhaka from 9–15 January 2021. Dhaka is the capital city in Bangladesh and a residence for more than 10.3 million people, or 6.29 percent of the total population of the country [37]. Between 8 March 2020 and 13 June 2021, there were 567,668 COVID-19 confirmed cases and 7,294 reported death in Dhaka [4]. While Dhaka makes up about 6.29% of the total population of Bangladesh, almost 69% of the COVID-19 cases in Bangladesh were reported in Dhaka as of 13 June (59.0% in Dhaka city alone) [4] making it one of the largest hotspots of COVID-19 [38].

### Study tool

A Barrier Analysis (BA) study was conducted to better understand the behavioral determinants of COVID-19 vaccine hesitancy in Dhaka. BA is a research tool that was developed in 1990 by Davis [39]. Based on HBM and TRA, BA studies explore respondents' beliefs about a behavior. Sometimes certain beliefs about a behavior (e.g., possible COVID-19 vaccination side effects) are common in a population, but are not necessarily associated with vaccine acceptance. BA is meant to identify the most likely behavioral determinants of a behavior [40]. A key feature of BA is that responses from those who have adopted a behavior or plan to ('Doers' or 'Acceptors') are compared with those who are have not or do not plan to ('Non-doers' or 'Non-acceptors') in order to identify behavioral determinants linked with a particular behavior (e.g., handwashing with soap, getting a vaccine). This enables practitioners to develop more effective behavior change messages and activities. BA has been used in more than 40% of low-to-middle-income countries and used extensively by World Vision and other organizations during both the Ebola [41] and COVID-19 pandemics [42]. The beliefs and other responses regarding behavioral determinants assessed during BA (see Table 1) are identified with a focus on the most actionable findings. The other details of BA approach can be found elsewhere [43–48]. There are BA studies in the peer-reviewed literature on exclusive breastfeeding [49], handwashing with soap at critical times [43], timely oral polio vaccination and agricultural extension behaviors [50], dietary salt reduction [51], transition from the lactational amenorrhea method to other modern family planning methods [39] and cervical cancer screening [52].

**Table 1. The Generic description of the determinants [40] and contextualization for the current study.**

| Name of determinant | Generic description | Contextualization for the current study. |
|---|---|---|
| Perceived self-efficacy | An individual's belief that he/she can do a particular behavior given his/her current knowledge, resources and skills. | We asked the respondents what might make it easier and what might make it difficult for them to get a COVID-19 vaccine if it was available to them in the coming month free of charge. |
| Perceived social norms | The perception that people important to an individual think that he/she should do the behavior (injunctive norms), and plan to do the behavior (descriptive norms). | We asked respondents:<br>• what portion of the people they know did they think would get a COVID-19 vaccine if was available to the community in the coming month free of charge;<br>• if their close family and friends would want them to get a COVID-19 vaccine;<br>• if their community and religious leaders would want them to get a COVID-19 vaccine;<br>• who would approve of them getting a COVID-19 vaccine;<br>• who would disapprove of COVID-19 vaccination;<br>• if they would get a COVID-19 vaccine (upon availability) if a doctor or nurse recommended it. |
| Perceived positive consequences | What positive things a person thinks will happen as a result of performing a behavior. | We asked respondents about the advantages of getting a COVID-19 vaccine. |
| Perceived negative consequences | The negative things a person thinks will happen as a result of performing a behavior. | We asked respondents about the disadvantages of getting a COVID-19 vaccine. |
| Access | The degree of availability (to a particular audience) of the needed facilities, services, or materials required to adopt a given behavior. | We asked the respondents how difficult it would be for them to get to the clinic where vaccines are normally offered. |
| Cues to action / reminders | The presence of reminders that help a person remember to do a particular behavior. | Questions on this possible determinant were not included in the current study as we did not believe it would be relevant at this stage of vaccine rollout. |
| Perceived susceptibility/risk | A person's perception of how vulnerable or at risk they feel vis-à-vis the problem or disease. | Respondents were asked what portion of people in their community have had COVID-19, how likely they thought it was that someone in their household would contract COVID-19, and how concerned they were about getting COVID-19. |
| Perceived severity | Belief that the problem or disease (which the behavior can prevent) is serious. | Respondents were asked how serious it would be if someone who lives in their household contracted COVID-19. |
| Perceived action efficacy | The belief that by practicing the behavior one will avoid the problem or disease; that the behavior is effective in preventing the problem or disease. | Respondents were asked if they were to get the COVID-19 vaccine, how likely would it be that they would get COVID-19 disease after that. |
| Perceived divine will | A person's belief that it is God's / Allah's or the gods' will (depending on their faith) for him/her to have the problem and/or to overcome it. | Respondents were asked if they thought that Allah / God / the gods approved or disapproved of people getting a COVID-19 vaccine. There were also asked if they agreed with the statement, *"Whether I get COVID-19 or not is purely a matter of God's will or chance. The actions I take will have little bearing on whether or not I get COVID-19."* |
| Policy | Laws and regulations (local, regional, or national) that affect adoption of the behavior and access to products and services. | Omitted from the study as the Government has decided to vaccinate its population and started vaccination program. |
| Culture | The set of history, customs, lifestyles, values, and practices within a self-defined group. | Respondents were asked if there were any cultural or religious reasons that they would not get a COVID-19 vaccine, and if yes, what those reasons were. |

## Questionnaire development

This study modified the standardized Barrier Analysis questionnaire from the *Designing for Behavior Change* (DBC) training manual [48].

The BA questionnaire is divided into three parts (Please see S1 File) In the first section, three questions were asked in order to categorize the potential respondents either an 'Acceptor' or a 'Non-acceptor' of COVID-19 vaccine. Specifically, we asked their age, and–if a COVID-19 vaccine was available to them in the coming months–how likely they would be to go for vaccination. The second section included four questions on their demographic

background, specifically on the respondents' age, gender, level of education and profession. The third section included determinant-specific questions. Based on the nature of determinant, both close-ended and open-ended questions were used to assess 10 of the usual 12 determinants of BA [40, 47]. Trust on vaccine information (provided by government officials, political, religious and community leaders), trust in vaccines, exposures to misinformation, and safety and efficacy of the vaccine are considered as important factors in vaccine acceptance in previous studies [8, 53–55], so questions on these factors were explored in this study, a well. Moreover, a previous study using BA method in the Ebola Vaccine Deployment and Compliance Project [41] found important insights by exploring these factors. In addition, based on local social media listening, one question was added to explore respondents' beliefs about herd immunity. The questionnaire was pretested among 12 respondents (6 acceptors and 6 non-acceptors) to check suitability of the language and slight modifications were made. (The responses from this pre-test were excluded from the current analysis.) After completing the pretest, modifications were made into Bengali version and those modifications were translated back into the English version.

## Sampling

The Barrier Analysis approach recommends a minimum sample size of 45 Doers (Acceptors) and 45 Non-doers (Non-acceptors) in order to detect statistically-significant Odds Ratios of 3.0 or higher with an alpha error of 5% and a power of 80% [40]. We interviewed adult men and women for this BA study and selected them through a convenience sampling strategy from six different areas of Dhaka, the capital city of Bangladesh. Enumerators from these six wards chose a starting household near where they lived, and then went door-to-door to identify respondents based on the questionnaire logic. Each enumerator was given a quota of 7 or 8 Acceptors and 7 or 8 Non-acceptor for a total of 15 respondents each.

## Data collection, management and analysis

We collected data from 9 to 13 January 2021 through individual interviews with responses recorded on paper-based questionnaires by three teams composed of one female and one male member. Male enumerators interviewed male respondents, and female enumerators interviewed female respondents. A research supervisor assured the quality of data. Following data collection, the data collection team and the lead author coded the open-ended responses thematically, using both an inductive and deductive coding process. At the end of this process, the team quantified the responses in each category for Acceptors and Non-acceptors separately. These categories and the number of responses registered for each were then entered into a standardized BA tabulation sheet that revealed whether differences in the proportion of Acceptors and Non-acceptors providing each response were statistically significant and should be addressed through the behavior change strategy. For each question and category of responses, the BA tabulation calculates the percentage of responses for both Acceptors and Non-Acceptors; the Odds Ratio, the Standard Error, and its confidence interval; the Estimated Relative Risk (ERR) [56]; and p-values (see S2 File). This allows practitioners to identify those differences between Acceptors and Non-acceptors that are statistically significant (at p<0.01) and to see the strength of the associations between each response and the behavior (based on the ERR).

## Ethical considerations

We performed all procedures in this study in accordance with the ethical standards of the institutional and national research committee and with the 1964 Helsinki Declaration and its later

amendments or comparable ethical standards. The study protocol was approved by the institutional Ethics Committee of the Chattogram Veterinary and Animal Sciences University, Bangladesh (permit ref. no. CVASU/Dir (R and E) EC/2020/169). We informed respondents about the study objectives, and obtained their written consent before conducted interview. The data collection activities were performed following the COVID-19 safety protocols in Bangladesh that were enacted by the Directorate General of Health Services in Bangladesh [57].

## Results

### Respondents' demographic profile

The characteristics of the study interviewees are shown in Fig 1. The majority of the respondents were male (71% and 58% of acceptors and Non-acceptors, respectively), and most of them belonged to the 18–25 years of age group (29% and 29% respectively). In terms of educational attainment, the majority of the respondents had completed education until 10th grade (36% and 49% of Acceptors and Non-acceptors, respectively) while most worked in services followed by small business.

### Determinant specific results

The statistically significant differences in responses and beliefs were found between Acceptors and Non-acceptors of COVID-19 vaccine are shown below. The categories of determinants are organized from higher to lower estimated relative risk (ERR). The detailed results are provided in the S2 File.

**Perceived social norms.** Some of the strongest predictors of vaccine acceptance in this population are beliefs around both injunctive and descriptive social norms: who the respondent thinks approves or disapproves of COVID-19 vaccination, and the proportion of people that they think will go for a COVID-19 vaccine when it is available. The results are shown in Table 2. Specifically, Acceptors were 3.2 times more likely to say they would be "very likely" to get a COVID-19 vaccine if a doctor or nurse approved (p<0.001), while Non-acceptors were 2.6 more likely to say it would be "not likely" that they would get a vaccine if a doctor or nurse recommended it (p<0.001). Acceptors were almost twice as likely to say that "most people" they know will get a vaccine (p<0.001), and 1.3 times more likely to say that "most close family and friends" will get a vaccine (p = 0.003). Conversely, Non-acceptors were 3.5 times more likely to say that "very few people" they knew would get a vaccine (p<0.001) and 1.3 times

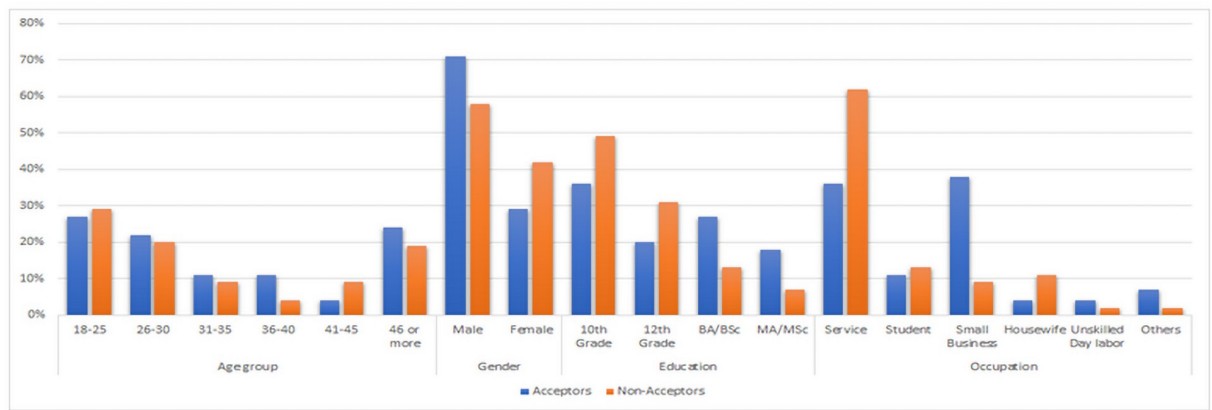

**Fig 1. Respondents' demographic profile.**

**Table 2. Perceived social norms.**

| Determinants/response | Doers n (%) | Non-Doers n (%) | Diff.* | Odds Ratio | 95% CI | ERR** | p-value |
|---|---|---|---|---|---|---|---|
| **Will you get a COVID-19 vaccine if doctor or nurse recommends?** | | | | | | | |
| Very likely | 42 (93) | 14 (31) | 62% | 31 | 8.19–117.28 | 3.18 | <0.001 |
| Somewhat likely | 3 (7) | 20 (44) | -38% | 0.09 | 0.02–0.33 | 0.44 | <0.001 |
| Not likely at all | 1 (2) | 9 (20) | -18% | 0.09 | 0.01–0.75 | 0.38 | 0.008 |
| **Proportion of people you know will get vaccine?** | | | | | | | |
| Most people would get the vaccine | 38 (84) | 12 (27) | 58% | 14.93 | 5.26–42.33 | 1.96 | <0.001 |
| Very few people would get it | 1 (2) | 13 (29) | -27% | 0.06 | 0.01–0.45 | 0.29 | <0.001 |
| **Will most of your close family and friends would want you to get a COVID-19 vaccine?** | | | | | | | |
| Yes | 32 (71) | 18 (40) | 31% | 3.69 | 1.53–8.89 | 1.31 | 0.003 |
| No | 10 (22) | 22 (49) | -27% | 0.3 | 0.12–0.75 | 0.76 | 0.007 |
| **Who disapproves to take a COVID-19 vaccine?** | | | | | | | |
| No one disapproves | 32 (71) | 3 (7) | 64% | 34.46 | 9.05–131.22 | 1.72 | <0.001 |
| **Who approves to take a COVID-19 vaccine?** | | | | | | | |
| Mother | 18 (40) | 5 (11) | 29% | 5.33 | 1.77–16.1 | 1.27 | 0.002 |
| The people who will not take the vaccine | 3 (7) | 13 (29) | -22% | 0.18 | 0.05–0.67 | 0.58 | 0.006 |
| Elderly member/relatives | 7 (16) | 18 (40) | -24% | 0.28 | 0.1–0.75 | 0.73 | 0.009 |
| The people who will not take the vaccine | 3 (7) | 13 (29) | -22% | 0.18 | 0.05–0.67 | 0.58 | 0.006 |
| **Will most of your community leaders and religious leaders want you to get a COVID-19 vaccine?** | | | | | | | |
| Yes | 35 (78) | 23 (51) | 27% | 3.35 | 1.34–8.35 | 1.31 | 0.007 |
| No. | 6 (13) | 19 (42) | -29% | 0.21 | 0.07–0.6 | 0.66 | 0.002 |

*Difference,

**Estimated Relative Risk.

more likely to say that "most of their close family and friends" would not get a COVID-19 vaccine (p = 0.003). In terms of respondents' impressions concerning who disapproves of their getting a COVID-19 vaccine, Acceptors were 1.7 times more likely (than Non-acceptors) to say that "no one" would disapprove (p<0.001). Non-acceptors were 1.5 times more likely to say that "my mother" (p<0.001), 1.4 times more likely to say "elderly people" (p = 0.009), and 1.7 times more likely to say "people who will not get the vaccine" (p = 0.006) would disapprove of their getting a COVID-19 vaccine. In addition, Acceptors were 1.3 times more likely to say their mother would approve of their getting a COVID-19 vaccine (p<0.001). With regards to community and religious leaders' influence on the decision to get a COVID-19 vaccine, Acceptors were 1.3 times more likely to say that "most community and religious leaders" would want them to get a vaccine (p = 0.007), while Non-acceptors were 1.5 times more likely to say that most community leaders and religious leaders would not want them to get a COVID-19 vaccine (p = 0.002).

**Perceived safety of COVID-19 vaccines.** As in many places in the world, concerns about the safety of COVID-19 vaccines are affecting COVID-19 vaccination acceptance in this population. When asked how safe the COVID-19 vaccines are, Non-acceptors were 1.8 times more likely to say that COVID-19 vaccines are "not safe at all" (p<0.001) while Acceptors were 1.4 times more likely to say that COVID-19 vaccines are "mostly safe" (p<0.001) (Table 3).

**Perceived self-efficacy.** The respondents were asked two open-ended questions to understand what they believe might make it easier or difficult to get a COVID-19 vaccine once it was available to them free of charge. From the results (Table 4), factors related to *how* and *where*

**Table 3. Perceived safety and risk.**

| Determinants/response | Doers n (%) | Non-Doers n (%) | Diff. | Odds Ratio | 95% CI | ERR | p-value |
|---|---|---|---|---|---|---|---|
| Safety: How safe would it be for you to get a COVID-19 vaccine? | | | | | | | |
| Not safe at all | 7 (16) | 28 (62) | -47% | 0.11 | 0.04–0.31 | 0.57 | <0.001 |
| Mostly safe | 26 (58) | 8 (18) | 40% | 6.33 | 2.41–16.6 | 1.36 | <0.001 |

the vaccine would be given affects intention to vaccinate, including whether or not proper COVID-19 social distancing and prevention measures are maintained during vaccination.

Concerning what might make it easier, Acceptors were 1.4 times more likely to say "if vaccination is provided by government health care centers or hospitals" (p<0.001), 1.2 times more likely to say "school-based vaccination centers" (p = 0.003), 1.3 times more likely to say "if the vaccines are provided by establishing kiosks" (p = 0.001) and 1.2 times more likely to say "if proper health and safety (COVID-19) protocols are maintained while giving the vaccine" (p = 0.006). Non-acceptors were 1.2 times more likely to say "when vaccines are given at home" (p = 0.010), and 1.7 times more likely to say "if the vaccine has no side effects" (p = 0.002) would make it easier for them to get a COVID-19 vaccine.

When asked what might make it difficult to get a COVID-19 vaccine, Acceptors were 1.3 times more likely to say either "if there are no health measures in the vaccination center due to overcrowding" or "risk of getting infected with COVID-19 while vaccinating" (p = 0.001), and 1.2 times more likely to say "if the vaccinator does not follow proper COVID-19 preventive measures" (p = 0.005) would make it difficult to get a COVID-19 vaccine. Non-acceptors were 2.4 times more likely to say "if the vaccine has severe side-effects" (p<0.001) and 2 time more likely to say "if there is bribery" (p = 0.004) would make it difficult to get a COVID-19 vaccine.

**Table 4. Perceived self-efficacy.**

| Determinants/response | Doers n (%) | Non-Doers n (%) | Diff. | Odds Ratio | 95% CI | ERR | p-value |
|---|---|---|---|---|---|---|---|
| Self-Efficacy: What would make it easier? | | | | | | | |
| If vaccine is provided by government health centers or hospitals | 23 (51) | 5 (11) | 40% | 8.36 | 2.79–25.08 | 1.36 | <0.001 |
| School-based vaccination centers | 21 (47) | 8 (18) | 29% | 4.05 | 1.55–10.60 | 1.25 | 0.003 |
| If the vaccines are provided by establishing kiosks | 18 (40) | 4 (9) | 31% | 6.83 | 2.08–22.40 | 1.29 | 0.001 |
| If proper health and safety (COVID-19) protocols are maintained while giving vaccine | 20 (44) | 8 (18) | 27% | 3.7 | 1.41–9.70 | 1.23 | 0.006 |
| If the vaccines are given at home | 15 (33) | 27 (60) | -27% | 0.33 | 0.14–0.79 | 0.8 | 0.010 |
| When the vaccines do not have any side effects | 4 (9) | 16 (36) | -27% | 0.18 | 0.05–0.58 | 0.6 | 0.002 |
| Self-Efficacy: What would make it Difficult? | | | | | | | |
| If there are no health measures in the vaccination center due to overcrowd (or risk of getting infected with COVID-19 while vaccinating). | 18 (40) | 4 (9) | 31% | 6.83 | 2.08–22.4 | 1.29 | 0.001 |
| If the vaccinator does not follow proper COVID-19 preventive measures | 19 (42) | 7 (16) | 27% | 3.97 | 1.46–10.8 | 1.24 | 0.005 |
| When the vaccine has severe side-effects | 3 (7) | 22 (49) | -42% | 0.07 | 0.02–0.28 | 0.41 | <0.001 |
| If there is bribery | 2 (4) | 12 (27) | -22% | 0.13 | 0.03–0.61 | 0.49 | 0.004 |

**Table 5. Perceived positive and negative consequences.**

| Determinants/response | Doers n (%) | Non-Doers n (%) | Diff. | Odds Ratio | 95% CI | ERR | p-value |
|---|---|---|---|---|---|---|---|
| **Perceived Positive Consequences (Advantages)** | | | | | | | |
| Reduce the risk of COVID infection | 30 (67) | 16 (36) | 31% | 3.63 | 1.52–8.65 | 1.29 | 0.003 |
| Children can start school again | 17 (38) | 5 (11) | 27% | 4.86 | 1.6–14.7 | 1.25 | 0.003 |
| Reduction in COVID-19-related costs (mask, hand sanitizer, detergents, primary medicine, COVID-19 test, etc). | 16 (36) | 3 (7) | 29% | 7.72 | 2.06–28.9 | 1.28 | 0.001 |
| We can attend social and cultural activities | 14 (31) | 3 (7) | 24% | 6.32 | 1.67–23.9 | 1.25 | 0.003 |
| New employment/income sources will be increased | 17 (38) | 3 (7) | 31% | 8.5 | 2.28–31.7 | 1.3 | <0.001 |
| Attend prayers in congregation | 16 (36) | 4 (9) | 27% | 5.66 | 1.71–18.7 | 1.26 | 0.002 |
| No positive outcome | 0 | 5 (11) | | | | | |
| **Perceived Negative Consequences (Disadvantages)** | | | | | | | |
| Weakness | 23 (51) | 9 (20) | 31% | 4.18 | 1.64–10.7 | 1.27 | 0.002 |
| Itching and skin problem | 17 (38) | 3 (7) | 31% | 8.5 | 2.28–31.7 | 1.3 | 0.000 |
| Life threatening side effects | 7 (16) | 24 (53) | -38% | 0.16 | 0.06–0.44 | 0.63 | 0.000 |
| Unknown/new diseases | 6 (13) | 18 (40) | -27% | 0.23 | 0.08–0.66 | 0.68 | 0.004 |
| Infertility | 3 (7) | 15 (33) | -27% | 0.14 | 0.04–0.54 | 0.54 | 0.001 |

**Perceived positive consequences and perceived negative consequences.** Respondents were also asked what the positive and negative consequences (e.g., advantages and disadvantages) would be of getting a COVID-19 vaccine (Table 6). Acceptors were more likely to mention (as advantages of COVID-19 vaccination) reducing the risk of COVID-19 infection and benefits related to livelihoods, and re-starting economic activities and getting back to normal life. Specifically (Table 5), Acceptors were 1.3 times more likely to say that reducing the risk of COVID-19 infection (p = 0.003), 1.3 times more likely to say "we can attend social and cultural activities" (p = 0.003), 1.2 times more likely to say "children can start school again" (p = 0.003), 1.3 times more likely to mention "reduction in COVID-19 related costs" (e.g. masks, hand sanitizer, tests; p<0.001), 1.3 times more likely to say "employment and income opportunities will be increased" (p<0.001), and 1.3 times more likely to say "attending prayers in congregation" (p<0.002) as advantages of getting a COVID-19 vaccine.

When asked about the negative consequences (disadvantages) of COVID-19 vaccination, Acceptors were 1.3 times more likely to mention mild side effects of vaccination such as "weakness" (p<0.002) and "itching and skin problems" p<0.001). Meanwhile, Non-acceptors were 1.6 times more likely (than Acceptors) to mention life-threatening side effects (p<0.001), 1.5 times more likely to say "unknown/new diseases" (p<0.004), and 1.9 times more likely to say "infertility" (p<0.001) as disadvantages of getting a COVID-19 vaccine.

**Perceived action efficacy.** Counter-intuitively, Acceptors were 1.3 times more likely to say they were "somewhat likely" to get COVID-19 once they were vaccinated against it (p<0.001) while Non-acceptors were 2.8 times more likely to say that they were "not likely at all" to get COVID-19 once one was vaccinated against it (p<0.001) (Table 6). Focus group discussions may be used at a later point in time to explore this finding. Related to herd immunity,

**Table 6. Perceived action efficacy and trust in COVID-19 vaccine.**

| Determinants/response | Doers n (%) | Non-Doers n (%) | Diff. | Odds Ratio | 95% CI | ERR | p-value |
|---|---|---|---|---|---|---|---|
| **Action Efficacy—Likelihood of getting COVID-19 after getting COVID-19 vaccine?** | | | | | | | |
| Somewhat likely | 22 (49) | 6 (13) | 36% | 6.22 | 2.2–17.6 | 1.31 | <0.001 |
| Not likely at all | 2 (4) | 19 (42) | -38% | 0.06 | 0.01–0.3 | 0.35 | <0.001 |
| **Perception on herd immunity: If one has been infected with COVID-19, vaccination with the COVID-19 vaccine is unnecessary.** | | | | | | | |
| Agree a little | 9 (20) | 20 (44) | -24% | 0.31 | 0.12–0.80 | 0.76 | 0.012 |
| **Most people will eventually get infected with COVID-19, so getting a COVID-19 vaccine is unnecessary.** | | | | | | | |
| Agree a little / Agree a lot | 8 (18) | 29 (64) | -47% | 0.12 | 0.04–0.32 | 0.60 | <0.001 |
| **Trust (in COVID-19 vaccine)—How much would you trust a COVID-19 vaccine?** | | | | | | | |
| Trust it a moderate amount | 19 (42) | 6 (13) | 29% | 4.75 | 1.67–13.5 | 1.26 | 0.002 |
| Trust it a lot | 18 (40) | 2 (4) | 36% | 14.33 | 3.08–66.7 | 1.34 | <0.001 |
| Do not trust it at all | 2 (4) | 14 (31) | -27% | 0.1 | 0.02–0.49 | 0.44 | 0.001 |
| Trust it a little | 6 (13) | 23 (51) | -38% | 0.15 | 0.05–0.42 | 0.6 | <0.001 |

respondents were asked if they agree or disagree with the statement "If one has been infected with COVID-19, vaccination with the COVID-19 vaccine is unnecessary." The Non-Acceptors were 1.8 times more likely to 'agree a little' with this statement. Related to perceived action efficacy, and to explore beliefs on herd immunity, respondents were asked if they agreed or disagreed with the statement "Most people will eventually get infected with COVID-19, so getting a COVID-19 vaccine is unnecessary." Non-acceptors were 1.7 times more likely to say that they "agree a little" and "agree a lot" with the statement (p<0.001) while Acceptors were 1.5 times more likely to "disagree a lot" with the statement (p<0.001).

**Trust in COVID-19 vaccines.** As expected, trust in COVID-19 vaccines is highly predictive of intended vaccine acceptance in Dhaka. Acceptors were twice as likely to say that they trust the COVID-19 vaccines "a lot" or a "moderate amount" (p<0.001). Conversely, Non-acceptors were 1.7 times more likely to say that they "trust them a little" (p<0.001) and 2.3 times more likely to say that they "do not trust [COVID-19 vaccines] at all" (p = 0.001) (Table 6).

**Perceived risk / susceptibility (to COVID-19).** Perceived risk of getting COVID-19 and the level of concern about getting COVID-19 also appeared to be highly predictive of intended vaccine acceptance in Dhaka. Acceptors were 1.4 times more likely to say they it was "very likely" that someone in their household would get COVID-19 over the next 3 months (p<0.001) while Non-acceptors were 1.3 times more likely to say that was only "somewhat likely" (p = 0.005). When respondents were asked how concerned they were about someone in their household getting COVID-19, Acceptors were 1.3 times more likely to say that they were "very concerned" (p = 0.002) while Non-acceptors were 1.7 times more likely to say that they were only "a little concerned" (p<0.001). Additionally, Non-acceptors were 1.7 times more likely to say that "very few people" have had COVID-19 in their communities (p<0.001) (Table 7).

**Perceived severity (of COVID-19).** The perceived severity of COVID-19 was also predictive of intended vaccine acceptance. Respondents were asked how serious it would be if they or someone else in their household got COVID-19. Acceptors were 1.3 times more likely (than Non-acceptors) to say that it would be "very serious" (p = 0.001) (Table 7).

**(Perceived) access.** Perceived difficultly in reaching clinics that normally provide vaccines was predictive of intended COVID-19 vaccine acceptance in Dhaka. Non-acceptors were 1.6 times more likely to say that it was "very difficult" to get to the facility that normally provides

**Table 7. Perceived risk/susceptibility to COVID-19, perceived severity of COVID-19, and perceived access.**

| Determinants/response | Doers n (%) | Non-Doers n (%) | Diff. | Odds Ratio | 95% CI | ERR | p-value |
|---|---|---|---|---|---|---|---|
| Perceived Risk / Susceptibility—Likelihood of someone in your household getting COVID-19 over next 3 months? | | | | | | | |
| Very likely | 25 (56) | 4 (9) | 47% | 12.81 | 3.92–41.83 | 1.43 | <0.001 |
| Somewhat likely | 11 (24) | 24 (53) | -29% | 0.28 | 0.12–0.69 | 0.76 | 0.005 |
| Perceived Risk / Susceptibility—How concerned are you about getting COVID-19? | | | | | | | |
| Very concerned | 22 (49) | 8 (18) | 31% | 4.42 | 1.69–11.6 | 1.27 | 0.002 |
| A little concerned | 5 (11) | 21 (47) | -36% | 0.14 | 0.05–0.43 | 0.57 | <0.001 |
| Perceived Risk / Susceptibility—Proportion of people in your community who have had C-19? | | | | | | | |
| Very few people. | 4 (9) | 17 (38) | -29% | 0.16 | 0.05–0.53 | 0.58 | 0.001 |
| Severity—How serious if someone in your HH got COVID-19? | | | | | | | |
| Very serious | 27 (60) | 12 (27) | 33% | 4.13 | 1.69–10.1 | 1.3 | 0.001 |
| Access—How difficult for you to get to the clinic where vaccines are normally offered? | | | | | | | |
| Very difficult | 8 (18) | 28 (62) | -44% | 0.13 | 0.05–0.35 | 0.61 | <0.001 |
| Somewhat difficult | 20 (44) | 5 (11) | 33% | 6.4 | 2.13–19.23 | 1.3 | <0.001 |

vaccines (p<0.001), while Acceptors were 1.3 times more likely to say it was "somewhat difficult" to get to that facility (p<0.001) (Table 7).

**Perceived divine will.** Personal agency and religious beliefs often come into play with vaccine acceptance. In this study (Table 8), we assessed personal agency around COVID-19 infection by asking respondents' degree of agreement or disagreement with the statement, "Whether I get COVID-19 or not is purely a matter of God's will or chance, the actions I take will have little bearing on whether or not I get COVID-19." Agreement with this statement was found to be predictive of vaccine acceptance. Specifically, Acceptors were 1.2 times more likely to say that they "disagree a lot" (p = 0.005) with this statement. We asked the respondents whether they believed that Allah, God, or the gods approves or disapproves of people getting COVID-19 vaccines. While 80% of Acceptors and 78% of Non-acceptors said that a deity approved of COVID-19 vaccinations, there were no statistically significant differences between Acceptors and Non-acceptors for this question (S2 File).

**Rumors/ culture.** Respondents were asked if there were any cultural or religious reasons that they would not get a COVID-19 vaccine (Table 8). Acceptors were 1.3 times more likely to say that there were no cultural or religious reasons they would not get a COVID-19 vaccine (p = 0.01), while Non-acceptors were 1.3 more likely to say that there were reasons (p = 0.002).

**Table 8. Perceived divine will and culture/rumors.**

| Determinants/response | Doers n (%) | Non-Doers n (%) | Diff. | Odds Ratio | 95% CI | ERR | p-value |
|---|---|---|---|---|---|---|---|
| Divine Will—Agree/disagree with "whether I get COVID-19 or not is purely a matter of God's will or chance. | | | | | | | |
| Disagree a lot | 26 (58) | 13 (29) | 29% | 3.37 | 1.4–8.08 | 1.25 | 0.005 |
| Culture—Any cultural or religious reasons you would not get COVID-19 vaccine? | | | | | | | |
| No | 30 (67) | 18 (40) | 27% | 3 | 1.27–7.09 | 1.25 | 0.010 |
| Yes | 11 (24) | 25 (56) | -31% | 0.26 | 0.11–0.64 | 0.74 | 0.002 |
| (If yes to Culture) What reasons | | | | | | | |
| Use of pork fat while making vaccine–Islam does not allow this. | 5 (11) | 19 (42) | -31% | 0.17 | 0.06–0.51 | 0.61 | 0.001 |
| Use of haram ingredients in the vaccine | 8 (18) | 21 (47) | -29% | 0.25 | 0.09–0.65 | 0.71 | 0.003 |
| If yes to (rumors) What would stop you or others from seeking the vaccine? | | | | | | | |
| Producers' hide and seek activities related to vaccine accuracy in the clinical test | 16 (36) | 4 (9) | 27% | 5.66 | 1.71–18.7 | 1.26 | 0.002 |

When asked what those reasons were, Non-Acceptors were 1.6 more likely to say that they had heard that 'the vaccines were made with pork fat which is not allowed (*haram*) by Islam' (p = 0.001) and 1.4 times more likely to say that 'vaccines were made with haram ingredients' (p = 0.003). Respondents were also asked if they had seen or heard of anything that would stop them or others from seeking to get a COVID-19 vaccine. If they said yes, they were asked a follow-up question on what would stop them or others from seeking the vaccine. Regarding this question, Acceptors were 1.3 more likely to say that producers' "hide and seek activities" related to vaccine accuracy in the clinical testing would stop them or their peers from getting a COVID-19 vaccine (p = 0.002). By hide and seek activities of vaccine producers, the respondents were referring to perceived misinformation and incomplete information being given on clinical trials and the process of developing a safe vaccine.

## Discussion

This Barrier Analysis study on intended acceptance of COVID-19 vaccines revealed important differences in responses and beliefs between Acceptors and Non-acceptors regarding behavioral determinants of vaccine acceptance in this urban setting of Bangladesh. One important finding is that even if one or two determinants or barriers are addressed, there are a multitude of important determinants and barriers that may affect vaccine acceptance, and deserve attention. Access, for instance, was among other determinants found to be important in our study in along with perceived social norms (found to be important in 75% of all BA studies in a recent review) and positive/negative consequences of the behavior (found to be important in 56% of all BA studies in that review) [58].

The largest potential predictor and behavioral driver of COVID-19 vaccine acceptance in this population, based on the associations seen between responses and vaccine acceptance, was perceived social norms. Perceived social norms, depending on the context, may hinder or inspire one to get a vaccine [59]. In this study, Acceptors were more likely (than Non-acceptors) to say that most people they know and most of their close family and friends will get the COVID-19 vaccine, that no one would disapprove of their getting a vaccine, and that they would be very likely to get a vaccine if a doctor or nurse recommended it. Similarly, one systematic review showed health care professionals are influential in promoting vaccinations [60], and our study confirmed this, as well. Acceptors were also more likely to say that most of their community and religious leaders will want them to get a vaccine.

Other beliefs, for example safety and trust, about the vaccines themselves were important, as well [61, 62]. In this study, Non-acceptors were much more likely to say that COVID-19 vaccines are "not safe at all" and to say that they only "trust them a little." Conversely, Acceptors were more likely to say that the COVID-19 vaccines are "mostly safe" and to say that they trust them "a lot."

In line with the Health Belief Model, beliefs about the disease itself were highly correlated with–an predictive of–vaccine acceptance [29]. Acceptors were much more likely (than Non-acceptors) to say it was "very likely" that someone in their household would get COVID-19 over the next three months and to be "very concerned" about getting COVID-19. Conversely, Non-acceptors were much more likely to say that it would be only "somewhat likely" that someone in their household would get COVID-19 and that "very few people" have had COVID-19 in their community. In alignment with what Patrick et al. [63] showed regarding perceived risk as structural feature of vaccine decision, Acceptors of this study were also more likely to believe that it would be "very serious" if someone in their household got COVID-19.

Perceived behavioral control (which is influenced by things that make it difficult or easy to perform the behavior) also influence vaccine uptake [34]. The barriers and enablers which

were mentioned more often by Acceptors provide clues as to ways to make it easier to boost acceptance. When asked what would make it easier to get a COVID-19 vaccine, Acceptors were more likely (than Non-acceptors) to mention providing vaccination through government health facilities, schools, and kiosks, and having vaccinators maintain proper COVID-19 health & safety protocols. Responding to the question about what would make it difficult to get a COVID-19 vaccine, Non-acceptors were much more likely say "when the vaccine has severe side effects." Personal agency also came into play: Acceptors were much more likely to say that they did not believe that getting COVID-19 was purely a matter of God's will or chance.

Aligned with other studies on vaccination uptake [64, 65], the results of this study showed that perceived effects of vaccines are a key factor in the vaccine decision. Acceptors named several positive consequences of getting a COVID-19 vaccine more often than Non-acceptors including (1) reduced the risk of Covid-19 infection, (2) being able to attend social and cultural activities, (3) children being able to start school again, (4) reduction in COVID-19 related costs, (5) increased employment and income opportunities, and (6) being able to attend prayers in a group setting. Conversely, Non-acceptors asked about negative consequences of getting a COVID-19 vaccine were more likely to mention (than Acceptors) (1) life-threatening side effects, (2) developing unknown / new diseases, and (3) becoming infertile as disadvantages that they would expect if they were to get a COVID-19 vaccine. Surprisingly, Acceptors were 1 more likely to say they were "somewhat likely" to get COVID-19 once they were vaccinated against it (p<0.001) while Non-acceptors were more likely to say that they were "not likely at all" to get COVID-19 if they were vaccinated.

Lastly, Non-acceptors were more likely to hold beliefs about herd immunity that could reduce acceptance, saying that they agree a little or a lot with the statement that "most people will eventually get infected with COVID-19, so getting a COVID-19 vaccine is unnecessary".

## Limitations

This study has a number of limitations. First, given that this study was only done in a limited urban area, the results should not be generalized to the rest of Bangladesh or other countries. The results are most generalizable to the six wards where interviews were conducted, but may be useful for other parts of Dhaka Second, by design (as a means to make the analysis easier for practitioners and the method replicable by more practitioners), the BA approach does not consider respondents' socio-economic information including level of income, living conditions, or other factors which may lead to some confounding or interaction of variables. Third, while the questionnaire was based on a standard questionnaire which has been used in hundreds of BA studies, and pretested with about 12 respondents, the questionnaire did not undergo formal reliability checks (e.g., inter-rater reliability). For the same reasons, probably, the analysis revealed wider confidence intervals. Finally, while many current studies recognize that there is a spectrum of acceptance between those who accept, those who are undecided or hesitant, and those who refuse, for the purposes of this study and for ease of analysis, we defined vaccine acceptance in a binary way.

## Implications for behavior change strategy

With these limitations aside, the study has a number of merits that make it useful in designing an integrated behavior change strategy to increase acceptance of COVID-19 vaccines. Aligned with WHO's Technical Advisory Group on Behavioral Insights and Sciences for Health recommendation on social and behavioral drivers on COVID-19 vaccination [36], our study identified important beliefs and responses associated with different determinants of COVID-19 vaccine acceptance among urban population in Bangladesh which could be valuable to

informing contextualized behavioral intervention and engagement strategies to support COVID-19 vaccination. For example, to increase perceived positive social norms, especially for those who found to be important influencers of this behavior (e.g., medical staff and mothers), videotaping individuals giving testimonials in each neighborhood on why they plan to get the vaccine, and distributing them over media is one possible approach to leverage social norms. Other activities can be used to make acceptance more visible (e.g., stickers on households that say, "We plan to vaccinate!" or lapel pins with the same message). To increase the perception that COVID-19 is serious (to address low perceived severity), testimonials by people who have lost or almost lost family members due to COVID-19 disease could be used. To address perceived divine will, religious leaders of all faiths could be assisted in creating sermon outlines on maintaining one's health (and linking that with COVID-19 vaccines), and supported in creating radio spots to promote COVID-19 vaccines. In addition to prevention of COVID-19, other positive consequences of immunization mentioned more often by Acceptors should be disseminated. While not repeating any misinformation, it will be important to provide clear information on the known minor risks of COVID-19 vaccination as a way to combat misinformation on side effects that were mentioned more often by Non-acceptors (e.g., life-threatening conditions, new diseases, infertility). Clear and detailed information on how vaccines are made and tested should be disseminated to counter misinformation (e.g., that vaccines are made with pork fat or other haram ingredients). Stakeholders should also take into account the findings on things that may make vaccination easier for people, such as providing the vaccine in schools and kiosks (in addition to government health facilities) and to assure that the population understands that proper COVID-19 health and safety protocols will be maintained in places where vaccines are given.

## Conclusion

This BA study has revealed a host of important behavioral determinants associated with and predictive of intended COVID-19 vaccine acceptance among the study population in Dhaka, Bangladesh. Particularly, perceived social norms, and beliefs about safety and trust in COVID-19 vaccines, perceived risk of getting COVID-19 and severity of the disease, perceived action efficacy, and personal agency are predictive of COVID-19 vaccination seeking among this study population. Findings on these potential behavioral drivers of COVID-19 vaccination acceptance should be used in the development of vaccination and communication plans. The study also has uncovered some important beliefs on the positive consequences from both Acceptors and Non-acceptors, which could be leveraged in developing behavior change messages. The results suggest that an integrated behavior change strategy, focused broadly on the behavioral determinants found to be associated with vaccine acceptance and hesitancy, needs to be incorporated into existing vaccination plans to increase the acceptance and uptake of COVID-19 vaccines and to end the pandemic.

## Supporting information

**S1 File. Questionnaire with consent form.**
(DOCX)

**S2 File. Tabulation sheet with data.**
(XLSX)

**S3 File.**
(PDF)

**S4 File.**
(PDF)

## Acknowledgments

The authors would like to acknowledge Chattogram Veterinary and Animal Sciences University for permitting us to conduct this study. Our entire research team are thankful to all the respondents who participated in the study. We would also like to thank the anonymous reviewers and academic editor for their thoughtful comments, suggestions and observations which improved the clarity and accuracy of our paper.

## Author Contributions

**Conceptualization:** Md. Abul Kalam, Thomas P. Davis, Jr.

**Data curation:** Md. Abul Kalam, Robert Kanwagi, Ariful Islam.

**Formal analysis:** Md. Abul Kalam, Thomas P. Davis, Jr., Shahanaj Shano, Md. Ariful Islam.

**Funding acquisition:** Mohammad Mahmudul Hassan.

**Investigation:** Md. Nasir Uddin.

**Methodology:** Md. Abul Kalam, Thomas P. Davis, Jr.

**Project administration:** Md. Abul Kalam, Shahanaj Shano, Md. Nasir Uddin, Mohammad Mahmudul Hassan.

**Software:** Thomas P. Davis, Jr.

**Supervision:** Heidi J. Larson.

**Validation:** Md. Abul Kalam, Thomas P. Davis, Jr., Heidi J. Larson.

**Writing – original draft:** Md. Abul Kalam, Shahanaj Shano.

**Writing – review & editing:** Md. Abul Kalam, Thomas P. Davis, Jr., Shahanaj Shano, Md. Nasir Uddin, Md. Ariful Islam, Robert Kanwagi, Ariful Islam, Mohammad Mahmudul Hassan, Heidi J. Larson.

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
