## [Decision Letter · Decision Letter 0]

27 May 2021

PONE-D-21-13638

Exploring the Behavioral Determinants of COVID-19 Vaccine Acceptance among an Urban Population in Bangladesh: Implications for Behavior Change Interventions.

PLOS ONE

Dear Dr. Md. Abul Kalam,

Thank you for submitting your manuscript to PLOS ONE. After careful consideration, we feel that it has merit but does not fully meet PLOS ONE’s publication criteria as it currently stands. Therefore, we invite you to submit a revised version of the manuscript that addresses the points raised during the review process.

This study is focusing of identifying the behavioral determinants of COVID-19 vaccine acceptance among an Urban Population in Dahka, Bangladesh in order to design a suitable strategy to increase the acceptance and uptake rate of vaccination in Bangladesh. Please note that your manuscript was reviewed by 5 experts in the field. There is consensus agreement that the idea of the article is interesting. Meanwhile, some of the reviewers identified many important problems in your manuscript and provided copious comments (enclosed).  Explanation and modification of the research question is indicated, especially 81% of urban people would get vaccinated when a COVID-19 vaccine is available. Restructure and refreshing of the methodology section is also indicated.  This piece of work could be better if it was done on a larger sample size. The work at this stage is considered as a pilot study. The presentation of the result as shown in the tables was clumsy and difficult to follow.

Please note that further language improvements and checking for plagiarism is also indicated. Consider revising the spelling, grammar, diction, and syntax throughout the manuscript for increased clarity. 

The manuscript could be greatly strengthened by considering editing according to the specific Reviewers’ comments.

Please submit your revised manuscript by July 1 2021 11:59PM. If you will need more time than this to complete your revisions, please reply to this message or contact the journal office at plosone@plos.org. Please include the following items when submitting your revised manuscript:

We look forward to receiving your revised manuscript.

Kind regards,

Ammal Mokhtar Metwally, Ph.D (MD)

Academic Editor

PLOS ONE

Journal Requirements:

2. Please address the following:

- Please include a copy of the questionnaire and interview guides used in this study, in both the original language and English, as Supporting Information.

- Please refrain from stating p values as 0.000 and use the format p<0.0001.

Reviewers' comments:

Reviewer's Responses to Questions

**Comments to the Author**

1. Is the manuscript technically sound, and do the data support the conclusions?

Reviewer #1: Yes

Reviewer #2: Yes

Reviewer #3: Yes

Reviewer #4: Yes

Reviewer #5: Partly

2. Has the statistical analysis been performed appropriately and rigorously? 

Reviewer #1: Yes

Reviewer #2: Yes

Reviewer #3: Yes

Reviewer #4: Yes

Reviewer #5: Yes

3. Have the authors made all data underlying the findings in their manuscript fully available?

Reviewer #1: Yes

Reviewer #2: Yes

Reviewer #3: Yes

Reviewer #4: Yes

Reviewer #5: Yes

4. Is the manuscript presented in an intelligible fashion and written in standard English?

Reviewer #1: Yes

Reviewer #2: Yes

Reviewer #3: Yes

Reviewer #4: Yes

Reviewer #5: Yes

5. Review Comments to the Author

Reviewer #1: 1. It might be good to put few major statistical numbers in the result section of the abstract for readers who might not have time to see all the detail results in the article or attracted by the results in the abstracts and want to go on the detail results.

2. The introduction part on the number of infected people presented is 14 million and as of May 7, 2021 this number has increased to 157,564,677 with 3.28 million deaths and should be revised in the paper with current updated figures.

3. Even though the barrier analysis method recommends 45 samples for each group, a number of cells in the analysis are having very low number almost less than 10 or in some cases 0 or 1 which might influence the results including a very wide confidence interval in some of statistically significant results. The authors should describe the reasons for taking a low sample size when accessing people for similar interviews was not a problem.

Reviewer #2: Thank you for referring this manuscript for revision. The idea is interesting. The article studies an urgent problem, investigating the behavioral determinants of COVID-19 vaccine acceptance among an Urban Population in Dahka, the Capital of Bangladesh. Authors believe that identifying these determinants will help in designing a suitable strategy to increase the acceptance and uptake rate of vaccination in Bangladesh. Individuals have several reasons for adopting or resisting certain behavior. It is important to identify these barriers or facilitators which influence somebody’s willingness to get health services.

Introduction of this manuscript demonstrating the size of the problem of hesitancy or refusing COVID-19 vaccination in different countries. However, on displaying the situation in Bangladesh authors reported that 81% of urban people would get vaccinated when a COVID-19 vaccine is available (9). So, why did authors choose to investigate the Urban population despite this high rate of acceptance among them?

I think authors must search for another reference showing the size of the problem in Bangladesh.

If population in rural areas was investigated, it might be more sounding.

At the end of introduction, the authors stated that their study included different societal structures (Line 107, page 4) which was not presented in data.

Subjects and methods: The paragraph from line (112-117, page 6) needs revision and rephrasing. The duration of the study is very short (one week).

Study tool: This section describing Barrier analysis is copied from a previous article. Applying a plagiarism checker is essential.

Questionnaire development: Authors stated that this study modified the standardized Barrier Analysis questionnaire. They did not mention these modifications. They omitted one determinant only from the 12 ones without explanation. These items could be summarized to the four essential determinants.

The paragraph from line (0- 12, page 9) is also copied from a previous literature (Determinants of COVID-19 Vaccine Acceptance in Six Lower- and Middle-Income Countries). Applying a plagiarism checker is essential.

Sample: sample size is small. Building an intervention plan requires a representative sample from different societal positions.

Authors did not demonstrate the age range nor the sex ratio of the recruited subjects.

Authors did not declare the type of random sampling technique.

It was better to summarize the questionnaire, increase the sample size and include some rural districts.

Results: This section is confusing. A lot of tables, with a lot of data, with several titles and subtitles. I think the lengthy questionnaire with many open-ended questions yielded in numerous data.

Characteristics of the studied participants must be presented in a separate table showing age categories, sex ratio, educational levels, occupation, and social class which are important variables affecting subject’s behavior.

Why did items related to Perceived Social Norms is divided in table 1 and table 4?

Could the number of tables be less or replaced by figures for some determinants?

I could not identify which beliefs were most highly associated with acceptance and non-acceptance with COVID-19 vaccine.

Could authors do a regression analysis to conclude the predictors of each behavior, whether acceptance or non-acceptance?

Discussion: is well written, organized, updated, and provided a good model to increase the rate of COVID-19 vaccine acceptance among Bangladesh population. This strategy could be applied in other developing countries.

Conclusion: is written in a general way and not concentrating on the main findings

References: authors may need to revise this section to correct incomplete references and delete repeated ones as number 15 and 19.

After careful consideration, I think that this manuscript will likely be suitable for publication if it is revised to address the points mentioned before (Increasing the sample size, including rural plus urban population, summarizing the open-ended questions, applying plagiarism checker and so on). Therefore, my decision is "Major Revision."

Reviewer #3: Thank you for studying this important global issue of acceptance or non-acceptance of proposed COVID-19 vaccines. However, I suggest you edit the sentences below for better understanding.

Materials and Methods:

These sentences are either long or complicated to understand due to the absence of adequate punctuation marks.

"There are BA studies in the peer-reviewed literature on exclusive breastfeeding (46) HWWS among internally displaced women in the Kurdistan region of Iraq (40) timely oral polio vaccination agricultural extension behaviors in India (47), dietary salt reduction in Nepal (48) transition from the lactational amenorrhea method to other modern family planning methods in Bangladesh (36) and cervical cancer screening in Senegal." Please check it.

- Questionaire development:

"The extent to which a person believes that it is Allah approves (or God or the gods’ will) for him/her to do the behavior (e.g. to get a COVID-19 vaccine)."

Results:

Perceived Positive Consequences and Perceived Negative Consequences:

"While reducing the reduced risk of infection was important, Acceptors were more likely to point out benefits related to livelihood and economic benefits and life getting back to normal." Please verifythe construction of this sentence.

Acknowledgments:

"Our entire research team also grateful to all the respondents for their kind cooperation during the interviews."

Reviewer #4: Dear Authors,

I found the manuscript interesting your study and results are well reported.

A sub-heading in the "materials and methods" section should clearly describe.

I think that possible weaknesses regarding the validity and reliability of the developed questionnaire should be discussed in the limitations.

Best wishes

Reviewer #5: This manuscript presents the results of a barrier analysis illuminating differences between vaccine hesitant and vaccine accepting individuals in Dhaka Bangladesh along a number of dimensions.

The results show that vaccine hesitant and acceptance respondents differ along a number of dimensions – some of them quite predictable (e.g. people who think the vaccines are safe or believe they have a higher risk of getting COVID-19 in the near future are more likely to be vaccine acceptant), and others more unique or illustrative of this specific context. To me, perhaps the most interesting findings are in Table 3 as they show that the vaccine hesitant would prefer different delivery mechanisms (e.g. given at home) than the vaccine acceptant 9who, for example, are much more open to receiving it at government health centers or school-based vaccination centers). This could be helpful in crafting outreach efforts to reach hesitant communities.

My main concerns are three-fold. First, the sheer range of items compared across the seven tables blunts the force of the most interesting findings. There’s so much here, that the most interesting and unique findings get lost. I would strongly encourage the authors to think about how to highlight the most important findings in the main text, and perhaps to include full tables with comparisons of all items in an appendix. This would also allow the discussion to focus more squarely on the similarities and differences between the main findings here and those of studies examining COVID-19 vaccine hesitancy in other contexts (a literature that grows every day). Rather than primarily recapitulating the findings in the Discussion, this section could more profitably engage related research.

Second, the paper often (though not always) uses causal language to describe the differences observed between vaccine acceptant and hesitant individuals as the sources, or determinants, or drivers of differences in willingness to vaccinate. This design does not allow causal claims. In fact, I would argue that for many of the dimensions examined, the causal arrow could run in the opposite direction (i.e. people don’t know much about the vaccines, but they know whether they are likely to take it and that preference influences their answers to at least some of the questions asked) or the willingness to vaccinate question and some of the other questions are measuring/tapping into the same thing (e.g. is trust in the vaccine really independent of vaccination intention and therefore a “key driver” of it? Or are both of these questions tapping into the same underlying concept?).

Finally, and a smaller point, I was trick by a result in Table 2 on p. 14. Is it correct that 16% of “doers” – that is subjects who said they would take the vaccine – thought the vaccine was “not safe at all”? To be sure, this is much lower than among the “non-doers,” but it nonetheless was very surprising. Is it worth digging into this a bit more?

6. PLOS authors have the option to publish the peer review history of their article (what does this mean?). If published, this will include your full peer review and any attached files.

Reviewer #1: **Yes: **Mengistu Asnake Kibret

Reviewer #2: No

Reviewer #3: No

Reviewer #4: No

Reviewer #5: No

---

## [Author Response · Author response to Decision Letter 0]

19 Jun 2021

June 19,2021

We are pleased to provide a revised manuscript entitled: Exploring the Behavioral Determinants of COVID-19 Vaccine Acceptance among an Urban Population in Bangladesh: Implications for Behavior Change Interventions. We thank the reviewers for their overall enthusiasm for the study and their constructive comments which have allowed us to significantly improve the manuscript. 

We have responded to each of their comments, as detailed below. Our manuscript has even more relevance at a time when the world is experiencing a severe pandemic, and there are lot of anxiety and tensions around vaccine’s efficacy and safety. Therefore, studying behavioral determinants on vaccine hesitancy/acceptance, this study provides insight that will behavior change strategy into vaccination policy across different countries including Bangladesh.

We look forward to the review of our revised manuscript and hope that it is now considered acceptable for publication in PLoS One. 

Sincerely,

Md. Abul Kalam.

Specific responses to academic editor and reviewers:

Academic editor

Thank you for submitting your manuscript to PLOS ONE. After careful consideration, we feel that it has merit but does not fully meet PLOS ONE’s publication criteria as it currently stands. Therefore, we invite you to submit a revised version of the manuscript that addresses the points raised during the review process.

Comment: This study is focusing of identifying the behavioral determinants of COVID-19 vaccine acceptance among an Urban Population in Dahka, Bangladesh in order to design a suitable strategy to increase the acceptance and uptake rate of vaccination in Bangladesh. Please note that your manuscript was reviewed by 5 experts in the field. There is consensus agreement that the idea of the article is interesting. Meanwhile, some of the reviewers identified many important problems in your manuscript and provided copious comments (enclosed). Explanation and modification of the research question is indicated, especially 81% of urban people would get vaccinated when a COVID-19 vaccine is available. Restructure and refreshing of the methodology section is also indicated. This piece of work could be better if it was done on a larger sample size. The work at this stage is considered as a pilot study. The presentation of the result as shown in the tables was clumsy and difficult to follow.

Response: Thank you so much for sending our draft to peer review and getting consensus from the respected reviewers to consider it for publication. The reviewers have pointed out our paper’s drawbacks critically and we really appreciate for their time and efforts. In the consecutive section, we have responded each reviewer’s comments and made necessary changes in the revised version. Along with other comments and suggestions, having a wider sample size would have included, however, as recommended by BA experts, we followed the recommended sample size. At this point, we are not able to include additional samples. As you commented, the current version could be considered as pilot study. Based on the reviewers’ suggestions and recommendations and considering the limitations, we might design another BA study by adopting a wider sample and including both rural and urban areas in future upon funding and resources. 

Comment: Please note that further language improvements and checking for plagiarism is also indicated. Consider revising the spelling, grammar, diction, and syntax throughout the manuscript for increased clarity. 

Response: Thanks so much for your suggestion. The draft has been reviewed by two co-authors who are native English speakers (one British and one American). We believe, the revised version meets the criteria of standard English. 

Comment: The manuscript could be greatly strengthened by considering editing according to the specific Reviewers’ comments.

Response: We have revised the draft based on comments and suggestions that made by the respected reviewers. 

Journal Requirements:

Response: Thank you so much. We have complied to the formatting guidelines of main body and author affiliations. 

2. Please address the following:

- Please include a copy of the questionnaire and interview guides used in this study, in both the original language and English, as Supporting Information.

Response: Thank you so much for reminding us on the requirements. We have uploaded the original and English version of the questionnaire along with consent form. 

- Please refrain from stating p values as 0.000 and use the format p<0.0001.

Response: Thank you so much for your suggestion. We have replaced 0.000 with p<0.0001 while reporting p values throughout the text. 

Reviewers' comments:

Reviewer's Responses to Questions

Comments to the Author

1. Is the manuscript technically sound, and do the data support the conclusions?

Reviewer #1: Yes

Reviewer #2: Yes

Reviewer #3: Yes

Reviewer #4: Yes

Reviewer #5: Partly

2. Has the statistical analysis been performed appropriately and rigorously?

Reviewer #1: Yes

Reviewer #2: Yes

Reviewer #3: Yes

Reviewer #4: Yes

Reviewer #5: Yes

3. Have the authors made all data underlying the findings in their manuscript fully available?

Reviewer #1: Yes

Reviewer #2: Yes

Reviewer #3: Yes

Reviewer #4: Yes

Reviewer #5: Yes

4. Is the manuscript presented in an intelligible fashion and written in standard English?

Reviewer #1: Yes

Reviewer #2: Yes

Reviewer #3: Yes

Reviewer #4: Yes

Reviewer #5: Yes

 

5. Review Comments to the Author

Reviewer #1: 

Comment: 1. It might be good to put few major statistical numbers in the result section of the abstract for readers who might not have time to see all the detail results in the article or attracted by the results in the abstracts and want to go on the detail results.

Response: Thank you so much for your suggestion. Since we have a number of questions under each determinant and found a number of significant results, we only picked the name of determinants rather than presenting the whole statistics. However, we have added statistics with some important findings. Please see line number 38-47 in the revised version. 

Comment: 2. The introduction part on the number of infected people presented is 14 million and as of May 7, 2021 this number has increased to 157,564,677 with 3.28 million deaths and should be revised in the paper with current updated figures.

Response: Thanks so much for pointing this issue. We put these statistics at the time of first submission. We have updated this number as of current re-submission date. We will continue to update based on the times of revision, if any further review arrives. 

Comment: 3. Even though the barrier analysis method recommends 45 samples for each group, a number of cells in the analysis are having very low number almost less than 10 or in some cases 0 or 1 which might influence the results including a very wide confidence interval in some of statistically significant results. The authors should describe the reasons for taking a low sample size when accessing people for similar interviews was not a problem.

Response: Thank you so much for raising this concern. The lower number of responses captured in the case of open-ended questions/responses. For the same reason, the analysis recorded a widened confidence interval. However, we have mentioned the sample size issue in the limitation section. Please check line number 488-489 in the revised version. 

 

Reviewer #2: 

Comment: Thank you for referring this manuscript for revision. The idea is interesting. The article studies an urgent problem, investigating the behavioral determinants of COVID-19 vaccine acceptance among an Urban Population in Dahka, the Capital of Bangladesh. Authors believe that identifying these determinants will help in designing a suitable strategy to increase the acceptance and uptake rate of vaccination in Bangladesh. Individuals have several reasons for adopting or resisting certain behavior. It is important to identify these barriers or facilitators which influence somebody’s willingness to get health services.

Response: Thank you so much for your appreciation on the merit our manuscript. Many thanks for your time and efforts that you made to review our paper. 

Comment: Introduction of this manuscript demonstrating the size of the problem of hesitancy or refusing COVID-19 vaccination in different countries. However, on displaying the situation in Bangladesh authors reported that 81% of urban people would get vaccinated when a COVID-19 vaccine is available (9). So, why did authors choose to investigate the Urban population despite this high rate of acceptance among them?

I think authors must search for another reference showing the size of the problem in Bangladesh.

Response: We really appreciate your concern. Ideally, the full background was not representative. We have now updated the research problem in the revised version by referring some peer reviewed published data. Please check line number 90-99 in the revised version. 

Comment: If population in rural areas was investigated, it might be more sounding.

Response: Thank you much for your comment. We do agree with your comment, however, by nature, the was intended to assess behavioral determinants urban population, so we did not include rural population. However, based on your suggestion, we may design another barrier analysis study at the rural setting with wider sample size, upon funding and other resources. 

Comment: At the end of introduction, the authors stated that their study included different societal structures (Line 107, page 4) which was not presented in data.

Response: Thank you so much for your concern. However, we presented respondents’ socio-economic background in the revised version. Please check line number 245-254. 

Comment: Subjects and methods: The paragraph from line (112-117, page 6) needs revision and rephrasing. 

Response: Thanks so much for your suggestion. We have rephrased these lines accordingly. Please check line number 137-139 in the revised version. 

Comment: The duration of the study is very short (one week).

Response: By nature, the BA is a rapid method. We conducted 90 interviews by recruiting 6 enumerators. Each of them conducted 3 interviews in a day and it took 5 days to reach the desired sample size. 

Comment: Study tool: This section describing Barrier analysis is copied from a previous article. Applying a plagiarism checker is essential.

Response: Thanks so much for your concern. The six-country paper was drafted by the same research group. There were some mis-timing between two papers and honestly, this draft was developed first and then we developed that six-country paper. While the current draft went through a rigorous edit internally, the six-country paper (was drafted by the same author group) that posted in a pre-print server ahead of submission of the current draft. However, we have revised this section accordingly. Please check the section as a whole. 

Comment: Questionnaire development: Authors stated that this study modified the standardized Barrier Analysis questionnaire. They did not mention these modifications. They omitted one determinant only from the 12 ones without explanation. These items could be summarized to the four essential determinants.

Response: Thanks so much for your concern. We have mentioned about the modifications and the reasons of omission of two determinants in the revised version. Please check Box 1 and line number 184-194 in the revised version. 

Comment: The paragraph from line (0- 12, page 9) is also copied from a previous literature (Determinants of COVID-19 Vaccine Acceptance in Six Lower- and Middle-Income Countries). Applying a plagiarism checker is essential.

Response: Thanks again for your concern. It happened for the same reasons that explained above. We have revised the full section as a whole. 

Comment: Sample: sample size is small. Building an intervention plan requires a representative sample from different societal positions.

Response: Thanks so much for your comment. As we followed BA recommended sample size, the current study included 90 respondents. However, we have mentioned this in the limitation section. 

Comment: Authors did not demonstrate the age range nor the sex ratio of the recruited subjects.

Response: Thanks so much for your concern. We have added a figure representing respondents’ demographic profile. Please check. 

Comment: Authors did not declare the type of random sampling technique.

Response: Thanks so much for your comment. In the sampling sub-section of the methodology, we describe this process, like randomly through a convenience sampling strategy and the data collection team approached adult men and women until they reached 90 respondents. Please check line number 188-190. 

Comment: It was better to summarize the questionnaire, increase the sample size and include some rural districts.

Response: Thank you for your valuable suggestion. As we mentioned earlier, the study was focused in urban areas, so adding new samples and rural areas is difficult to accommodate at this point. But we honestly appreciate your suggestion and based on that we may design a follow up study including both rural and urban population with wider sample size in future, upon funding and other resources. 

Comment: Results: This section is confusing. A lot of tables, with a lot of data, with several titles and subtitles. I think the lengthy questionnaire with many open-ended questions yielded in numerous data.

Response: Thank you for your comment. We attempted to use figure, but unfortunately, we cannot show ERR, CI and OR for each item while showing the results in a figure. We have changed all the tables to show significant results only. Please see the result section as a whole. The other results are shown in Supplementary file 2. 

Comment: Characteristics of the studied participants must be presented in a separate table showing age categories, sex ratio, educational levels, occupation, and social class which are important variables affecting subject’s behavior.

Response: Thank you for your valuable suggestion. We have added a figure (figure 1) to represent demographic information of the respondents. 

Comment: Why did items related to Perceived Social Norms is divided in table 1 and table 4? Could the number of tables be less or replaced by figures for some determinants? I could not identify which beliefs were most highly associated with acceptance and non-acceptance with COVID-19 vaccine.

Response: Thanks so much for your close look. It was unintentional. We deleted these repetitive results. 

Comment: Could authors do a regression analysis to conclude the predictors of each behavior, whether acceptance or non-acceptance?

Response: Thank you so much for your suggestion. As we followed standard BA approach, we did not do a regression analysis, but we considered estimated relative risk of each response that captured from both acceptors and non-acceptors in a comparative manner. 

Comment: Discussion: is well written, organized, updated, and provided a good model to increase the rate of COVID-19 vaccine acceptance among Bangladesh population. This strategy could be applied in other developing countries.

Response: Thank you so much for your appreciation. 

Comment: Conclusion: is written in a general way and not concentrating on the main findings

Response: Thank you so much for your valuable suggestion. As we have a separate section on recommendation based on specific findings, in the conclusion, we urged to accommodate these recommendations into policy. However, we have revised this section. Please check the conclusion section. 

Comment: References: authors may need to revise this section to correct incomplete references and delete repeated ones as number 15 and 19.

Response: Thanks so much. We have made correction accordingly. 

Comment: After careful consideration, I think that this manuscript will likely be suitable for publication if it is revised to address the points mentioned before (Increasing the sample size, including rural plus urban population, summarizing the open-ended questions, applying plagiarism checker and so on). Therefore, my decision is "Major Revision."

Response: Finally, we really appreciate your rigorous comments, suggestions and concerns. We have addressed all of your comments and suggestions. We believe the revised version would be suitable for acceptance.  

Reviewer #3: Thank you for studying this important global issue of acceptance or non-acceptance of proposed COVID-19 vaccines. However, I suggest you edit the sentences below for better understanding.

Comment: Materials and Methods:

These sentences are either long or complicated to understand due to the absence of adequate punctuation marks.

"There are BA studies in the peer-reviewed literature on exclusive breastfeeding (46) HWWS among internally displaced women in the Kurdistan region of Iraq (40) timely oral polio vaccination agricultural extension behaviors in India (47), dietary salt reduction in Nepal (48) transition from the lactational amenorrhea method to other modern family planning methods in Bangladesh (36) and cervical cancer screening in Senegal." Please check it.

Response: Thank you so much for your suggestion. We have checked and made necessary changes. Please check line 158-163 in the revised version. 

Comment: - Questionaire development:

"The extent to which a person believes that it is Allah approves (or God or the gods’ will) for him/her to do the behavior (e.g. to get a COVID-19 vaccine)."

Response: Thanks so much for your concern. As per definition that provided in the BA module, this determinant assesses the belief on God’s will on the problem or solution. We adopted it from in our study to assess the level of belief. This modification has been mentioned in Box 1. 

Comment: Results:

Perceived Positive Consequences and Perceived Negative Consequences:

"While reducing the reduced risk of infection was important, Acceptors were more likely to point out benefits related to livelihood and economic benefits and life getting back to normal." Please verifythe construction of this sentence.

Response: Thanks so much for pointing out this, we have changed the table as whole in the revised version. Please check. 

Comment: Acknowledgments:

"Our entire research team also grateful to all the respondents for their kind cooperation during the interviews."

Response: Thanks so much for pointing out this. We have checked and made necessary changes. Please check line number 537-543.  

Reviewer #4: 

Comment: Dear Authors, I found the manuscript interesting your study and results are well reported.

Response: Thanks so much for your comment. We are so thankful for your time and efforts to read the draft. 

Comment: A sub-heading in the "materials and methods" section should clearly describe.

Best wishes

Response: Thanks so much for your suggestion. This heading is there in the revised version. 

Comment: I think that possible weaknesses regarding the validity and reliability of the developed questionnaire should be discussed in the limitations.

Best wishes

Response: Thanks so much for your suggestion. We have mentioned this in the limitation section.  

Reviewer #5: 

Comment: This manuscript presents the results of a barrier analysis illuminating differences between vaccine hesitant and vaccine accepting individuals in Dhaka Bangladesh along a number of dimensions.

The results show that vaccine hesitant and acceptance respondents differ along a number of dimensions – some of them quite predictable (e.g. people who think the vaccines are safe or believe they have a higher risk of getting COVID-19 in the near future are more likely to be vaccine acceptant), and others more unique or illustrative of this specific context. To me, perhaps the most interesting findings are in Table 3 as they show that the vaccine hesitant would prefer different delivery mechanisms (e.g. given at home) than the vaccine acceptant 9who, for example, are much more open to receiving it at government health centers or school-based vaccination centers). This could be helpful in crafting outreach efforts to reach hesitant communities. 

Response: Thanks so much for close observation and appreciation on the results. 

Comment: My main concerns are three-fold. First, the sheer range of items compared across the seven tables blunts the force of the most interesting findings. There’s so much here, that the most interesting and unique findings get lost. I would strongly encourage the authors to think about how to highlight the most important findings in the main text, and perhaps to include full tables with comparisons of all items in an appendix. This would also allow the discussion to focus more squarely on the similarities and differences between the main findings here and those of studies examining COVID-19 vaccine hesitancy in other contexts (a literature that grows every day). Rather than primarily recapitulating the findings in the Discussion, this section could more profitably engage related research.

Response: Thank you so much for your concern. We have now re-organised the result section, particularly, the tables. Specifically, we omitted the insignificant results from the tables and referred them to the supplementary file 2. Hope the result section is clearer. 

Comment: Second, the paper often (though not always) uses causal language to describe the differences observed between vaccine acceptant and hesitant individuals as the sources, or determinants, or drivers of differences in willingness to vaccinate. This design does not allow causal claims. In fact, I would argue that for many of the dimensions examined, the causal arrow could run in the opposite direction (i.e. people don’t know much about the vaccines, but they know whether they are likely to take it and that preference influences their answers to at least some of the questions asked) or the willingness to vaccinate question and some of the other questions are measuring/tapping into the same thing (e.g. is trust in the vaccine really independent of vaccination intention and therefore a “key driver” of it? Or are both of these questions tapping into the same underlying concept?).

Response: Thank you so much for your concern. This is really an important observation. Concerning this, we have changed the language throughout, only referring to "drivers" of vaccine acceptance/hesitancy when referring to other documents (by the WHO) that use that language. We now talk more about determinants and their "correlation with vaccine acceptance" and "predictors of vaccine acceptance.

Comment: Finally, and a smaller point, I was trick by a result in Table 2 on p. 14. Is it correct that 16% of “doers” – that is subjects who said they would take the vaccine – thought the vaccine was “not safe at all”? To be sure, this is much lower than among the “non-doers,” but it nonetheless was very surprising. Is it worth digging into this a bit more?

Response: Thanks so much for pointing out this issue. We are also concerned about this. As the current study unable to dig further at this point, we could design another studies to understand the changes and dig out this issue in future, upon funding and other resources. 

6. PLOS authors have the option to publish the peer review history of their article (what does this mean?). If published, this will include your full peer review and any attached files.

Do you want your identity to be public for this peer review? For information about this choice, including consent withdrawal, please see our Privacy Policy.

Reviewer #1: Yes: Mengistu Asnake Kibret

Reviewer #2: No

Reviewer #3: No

Reviewer #4: No

Reviewer #5: No

---

## [Decision Letter · Decision Letter 1]

22 Jul 2021

PONE-D-21-13638R1

Exploring the Behavioral Determinants of COVID-19 Vaccine Acceptance among an Urban Population in Bangladesh: Implications for Behavior Change Interventions.

PLOS ONE

Dear Dr. Kalam,

Thank you for submitting your manuscript to PLOS ONE. After careful consideration, we feel that it has merit but does not fully meet PLOS ONE’s publication criteria as it currently stands. Therefore, we invite you to submit a revised version of the manuscript that addresses the points raised during the review process.

Before accepting your article, you have to upload your data 

We look forward to receiving your revised manuscript.

Kind regards,

Ammal Mokhtar Metwally, Ph.D (MD)

Academic Editor

PLOS ONE

Journal Requirements:

Reviewers' comments:

Reviewer's Responses to Questions

**Comments to the Author**

1. If the authors have adequately addressed your comments raised in a previous round of review and you feel that this manuscript is now acceptable for publication, you may indicate that here to bypass the “Comments to the Author” section, enter your conflict of interest statement in the “Confidential to Editor” section, and submit your "Accept" recommendation.

Reviewer #3: All comments have been addressed

Reviewer #5: All comments have been addressed

2. Is the manuscript technically sound, and do the data support the conclusions?

Reviewer #3: Yes

Reviewer #5: Yes

3. Has the statistical analysis been performed appropriately and rigorously? 

Reviewer #3: Yes

Reviewer #5: Yes

4. Have the authors made all data underlying the findings in their manuscript fully available?

Reviewer #3: Yes

Reviewer #5: Yes

5. Is the manuscript presented in an intelligible fashion and written in standard English?

Reviewer #3: Yes

Reviewer #5: Yes

6. Review Comments to the Author

Reviewer #3: (No Response)

Reviewer #5: Thank you for addressing my concerns. I am now pleased to support this paper's publication in PLOS One.

7. PLOS authors have the option to publish the peer review history of their article (what does this mean?). If published, this will include your full peer review and any attached files.

Reviewer #3: No

Reviewer #5: No

---

## [Author Response · Author response to Decision Letter 1]

30 Jul 2021

July 28, 2021

We are pleased to provide a revised manuscript entitled: Exploring the Behavioral Determinants of COVID-19 Vaccine Acceptance among an Urban Population in Bangladesh: Implications for Behavior Change Interventions. In this round, we do not have specific comments from the respected reviewers and they declared that we have addressed their comments in previous round. The only comment from the academic editor was on the data availability, which has been already uploaded in the system during revision round 1. However, we have made corrections on the manuscript, mostly on grammatical issues and choice of words – as part of proof-reading. This was done by a native English speaker who is also a co-author. 

We look forward to the review of our revised manuscript and hope that it is now considered acceptable for publication in PLoS One. 

Sincerely,

Md. Abul Kalam.

Specific responses to academic editor and reviewers:

Thank you for submitting your manuscript to PLOS ONE. After careful consideration, we feel that it has merit but does not fully meet PLOS ONE’s publication criteria as it currently stands. Therefore, we invite you to submit a revised version of the manuscript that addresses the points raised during the review process.

Response: Thank you so much for considering our manuscript for publication. We have updated the draft based on suggestions and recommendations. 

Comment: Concerning Data Availability, you have declared that all data are fully available without restriction 

Before accepting your article, you have to upload your data. 

Response: Thank you so much for your concern. We uploaded the data file named “S2 File: Tabulation sheet with data.” This file can be found under the "File Inventory" tab. 

Comment: Response: Thanks so much for your suggestion. The financial disclosure is final as it was during the first submission. 

Comment: If applicable, we recommend that you deposit your laboratory protocols in protocols.io to enhance the reproducibility of your results. Protocols.io assigns your protocol its own identifier (DOI) so that it can be cited independently in the future. For instructions see: http://journals.plos.org/plosone/s/submission-guidelines#loc-laboratory-protocols. Additionally, PLOS ONE offers an option for publishing peer-reviewed Lab Protocol articles, which describe protocols hosted on protocols.io. Read more information on sharing protocols at https://plos.org/protocols?utm_medium=editorial-email&utm_source=authorletters&utm_campaign=protocols.

Response: Thanks so much for your suggestion. This is Not Applicable for us. 

Journal Requirements:

Response: Thanks so much for your suggestion. We have replaced one reference (no. 12) that was published in the national newspaper which is now available online. 

Reviewers' comments:

Reviewer's Responses to Questions

Comments to the Author

1. If the authors have adequately addressed your comments raised in a previous round of review and you feel that this manuscript is now acceptable for publication, you may indicate that here to bypass the “Comments to the Author” section, enter your conflict-of-interest statement in the “Confidential to Editor” section, and submit your "Accept" recommendation.

Reviewer #3: All comments have been addressed

Reviewer #5: All comments have been addressed

2. Is the manuscript technically sound, and do the data support the conclusions?

Reviewer #3: Yes

Reviewer #5: Yes

3. Has the statistical analysis been performed appropriately and rigorously?

Reviewer #3: Yes

Reviewer #5: Yes

4. Have the authors made all data underlying the findings in their manuscript fully available?

Reviewer #3: Yes

Reviewer #5: Yes

5. Is the manuscript presented in an intelligible fashion and written in standard English?

Reviewer #3: Yes

Reviewer #5: Yes

6. Review Comments to the Author

Reviewer #3: (No Response)

Reviewer #5: Thank you for addressing my concerns. I am now pleased to support this paper's publication in PLOS One.

7. PLOS authors have the option to publish the peer review history of their article (what does this mean?). If published, this will include your full peer review and any attached files.

Do you want your identity to be public for this peer review? For information about this choice, including consent withdrawal, please see our Privacy Policy.

Reviewer #3: No

Reviewer #5: No

Comment: Response: Thanks so much for your suggestion. We addressed these comments during the revision Round 1. 

Comment: While revising your submission, please upload your figure files to the Preflight Analysis and Conversion Engine (PACE) digital diagnostic tool, https://pacev2.apexcovantage.com/. PACE helps ensure that figures meet PLOS requirements. To use PACE, you must first register as a user. Registration is free. Then, login and navigate to the UPLOAD tab, where you will find detailed instructions on how to use the tool. If you encounter any issues or have any questions when using PACE, please email PLOS at figures@plos.org. Please note that Supporting Information files do not need this step.

Response: Thanks so much for your suggestion. We have one figure which was formatted by PACE and uploaded during Revision round 1. The figure can be found in the File Inventory Tab.

---

## [Decision Letter · Decision Letter 2]

9 Aug 2021

Exploring the Behavioral Determinants of COVID-19 Vaccine Acceptance among an Urban Population in Bangladesh: Implications for Behavior Change Interventions.

PONE-D-21-13638R2

Dear Dr. Kalam,

We’re pleased to inform you that your manuscript has been judged scientifically suitable for publication and will be formally accepted for publication once it meets all outstanding technical requirements.

Kind regards,

Ammal Mokhtar Metwally, Ph.D (MD)

Academic Editor

PLOS ONE

Additional Editor Comments (optional):

Great effort was made by the authors to utilize the feedback that was provided for them to correct for resubmission and all comments have been addressed.

The corresponding author declared that all data are fully available without restriction. Accordingly, the raw data are requested to be uploaded as per PLOS one publications requirements for the manuscript to be published. 

Reviewers' comments:

Reviewer's Responses to Questions

**Comments to the Author**

1. If the authors have adequately addressed your comments raised in a previous round of review and you feel that this manuscript is now acceptable for publication, you may indicate that here to bypass the “Comments to the Author” section, enter your conflict of interest statement in the “Confidential to Editor” section, and submit your "Accept" recommendation.

Reviewer #3: All comments have been addressed

Reviewer #5: All comments have been addressed

2. Is the manuscript technically sound, and do the data support the conclusions?

Reviewer #3: Yes

Reviewer #5: Yes

3. Has the statistical analysis been performed appropriately and rigorously? 

Reviewer #3: Yes

Reviewer #5: Yes

4. Have the authors made all data underlying the findings in their manuscript fully available?

Reviewer #3: Yes

Reviewer #5: Yes

5. Is the manuscript presented in an intelligible fashion and written in standard English?

Reviewer #3: Yes

Reviewer #5: Yes

6. Review Comments to the Author

Reviewer #3: (No Response)

Reviewer #5: (No Response)

7. PLOS authors have the option to publish the peer review history of their article (what does this mean?). If published, this will include your full peer review and any attached files.

Reviewer #3: No

Reviewer #5: No

---

## [Editor Report · Acceptance letter]

12 Aug 2021

PONE-D-21-13638R2 

Exploring the Behavioral Determinants of COVID-19 Vaccine Acceptance among an Urban Population in Bangladesh: Implications for Behavior Change Interventions. 

Dear Dr. Kalam:

I'm pleased to inform you that your manuscript has been deemed suitable for publication in PLOS ONE. Congratulations! Your manuscript is now with our production department. 

Kind regards, 

on behalf of

Professor Ammal Mokhtar Metwally 

Academic Editor

PLOS ONE